# A Comparative Study Evaluating the Effectiveness of Folate-Based B Vitamin Intervention on Cognitive Function of Older Adults under Mandatory Folic Acid Fortification Policy: A Systematic Review and Meta-Analysis of Randomized Controlled Trials

**DOI:** 10.3390/nu16142199

**Published:** 2024-07-10

**Authors:** Liyang Zhang, Xukun Chen, Yongjie Chen, Jing Yan, Guowei Huang, Wen Li

**Affiliations:** 1Department of Nutrition and Food Science, School of Public Health, Tianjin Medical University, 22 Qixiangtai Road, Heping District, Tianjin 300070, China; zly3485@tmu.edu.cn (L.Z.); chenxk1209@tmu.edu.cn (X.C.); huangguowei@tmu.edu.cn (G.H.); 2Department of Epidemiology & Biostatistics, School of Public Health, Tianjin Medical University, Tianjin 300070, China; chenyongjie@tmu.edu.cn; 3Tianjin Key Laboratory of Environment, Nutrition and Public Health, Tianjin 300070, China; yanjing@tmu.edu.cn; 4Department of Social Medicine and Health Administration, School of Public Health, Tianjin Medical University, Tianjin 300070, China

**Keywords:** folic acid, cognitive function, mandatory fortification policy, older adults, meta-analysis

## Abstract

The policies regarding the mandatory fortification of food with folic acid (FA) may impact the effectiveness of folate-based B vitamin treatment on cognitive function in older adults. We critically and systematically review the literature to assess whether food fortification policies affect folate-based B vitamin treatment efficacy on cognition function in older adults. Electronic databases, including PubMed, Web of Science, and CNKI, were searched for “Cognitive Function”, “Folate”, and “Older Adults”. The study had specific criteria for inclusion, which were as follows: (1) the studies should initially have randomized controlled trials that were conducted on older adults aged 60 or above; (2) the studies must assess the relationship between folate status and cognitive performance; and (3) the studies should clarify the policies regarding food fortification with FA. This review followed the Preferred Reporting Items for Systematic Reviews and Meta-Analyses (PRISMA) reporting guidelines. Two reviewers independently extracted all the data, and any discrepancies were resolved by consensus. All the data collected were compiled, compared, and analyzed critically. Random effects models were used to assess the effects of interventions. The systematic review included fifty-one articles involving 42,768 participants. Of these, the 23 articles were included in the meta-analysis. The meta-analysis on the effects of folate-based B vitamin supplementation on cognitive function showed a significant overall impact (Z = 3.84; *p* = 0.0001; SMD, 0.18; 95% CI, 0.09, 0.28). Further analysis revealed that FA food fortification policies were not implemented in countries where folate-based B vitamin supplementation improved cognitive impairment in older adults (Z = 3.75; *p* = 0.0002; SMD, 0.27; 95% CI, 0.13, 0.40). However, the FA intervention did not have significant outcomes in areas where FA food fortification policies were mandatory (Z = 0.75; *p* = 0.45; SMD, 0.03; 95% CI, −0.06, 0.13). Supplementing with oral folic acid, alone or in combination, has been linked to improved cognitive performance in older adults. While mandatory FA fortification has the improved folic acid status, additional folate-based B vitamin supplements do not appear to influence cognitive function.

## 1. Introduction

Several recent studies have explored the connection between B vitamin levels and cognitive function in older adults [1,2]. Folic acid (FA), vitamin B9, plays a crucial role throughout life [3]. Various published studies have investigated the role of folic acid in preventing cognitive decline in adults. For example, an analysis of 11,276 participants in a prospective Brazilian cohort found that folic acid, naturally found in food, could help moderate cognitive decline [4]. However, a study of 466,224 participants in the UK Biobank cohort found that folic acid supplementation alone was linked to an increased risk of Alzheimer’s disease [5]. Furthermore, the results from meta-analysis have been conflicting, leading to inconclusive findings [6,7]. Therefore, the most recent study’s results could be inconsistent, and no clear conclusion exists. 

Different countries or regions follow varying FA fortification policies, which leads to a difference in the folate nutritional status of the population. Due to inadequate intake, folate deficiency is common among older adults in countries or regions that do not follow mandatory FA fortification policies [8]. However, the rates of folate deficiency are significantly lower among older populations in countries where the FA food fortification is mandatory. For instance, Brazil made FA supplementation mandatory in wheat and corn flour in 2004, resulting in folate deficiency rates of only 2.09% among people above 60 years of age [9].

It is essential to investigate whether the different nutritional statuses of FA in the population, as a result of diverse food fortification policies, affect the impact of FA supplementation on cognitive function in older adults. Therefore, a meta-analysis of randomized controlled trials (RCTs) is necessary to provide a comprehensive and up-to-data summary of the effect of FA supplementation on cognitive performance among older adults, whether they live in countries with or without mandatory FA food fortification.

## 2. Methods

### 2.1. Protocol and Registration

This protocol was submitted to the PROSPERO register (CRD42023439711) and is available at: https://www.crd.york.ac.uk/PROSPERO/ (accessed on 28 June 2023), which was reported according to the Preferred Reporting Items for Systematic Reviews and Meta-Analyses (PRISMA) guideline [10].

### 2.2. Eligibility Criteria

The research question was “assess whether policies of FA food fortification affect the efficacy of FA treatment on cognition function in older adults”, according to the Population, Intervention/Exposure, Comparison, Outcomes and Study design (Table 1). The study’s inclusion criteria were as follows: original published studies with intervention design (RCTs) conducted in older adults (aged 60 or above). The study assessed the relationship between folate status and cognitive performance. The study should clarify the policies regarding food fortification with FA. Excluded from the study were theses, dissertations, monographs, case reports, annals, and literature reviews (narrative, integrative, systematic, and meta-analysis). Additionally, participants with depression, psychosis, or Parkinson’s disease were omitted. Studies needing more data on the variation of cognitive scale measures have also been eliminated. No restrictions were set concerning race, origin, publication year, or follow-up length.

### 2.3. Information Sources

Electronic databases, including PubMed, Web of Science, and CNKI, were searched for literature up to June 2023. 

### 2.4. Search Strategy

The descriptors were defined from the Health Sciences Descriptors (DeCS), in English, and combined using the Boolean operators OR and AND. The search was based on the following search strategy: (folic acid OR folate OR folic acid supplementation OR folate supplementation) AND (cognition OR cognition decline OR cognitive impairment), while limiting the types of studies to RCTs. The last search was performed on 25 June, Reviewing the reference lists of published systematic reviews that met our inclusion criteria further supplemented the search. 

### 2.5. Selection Process

Two researchers conducted an independent evaluation of the retrieved citations during the study selection process. The EndNote was used to exclude duplicates and screen the titles and abstracts according to the eligibility criteria in phase 2, relevant articles were selected for full-text reading. Any disagreements during the phase 1 evaluation process were resolved, and the article was included for the next step. In case of disagreements after the whole reading in phase 2, the reviewers discussed and resolved the issue. If necessary, a third researcher (Li. W.) was consulted. 

### 2.6. Data Collection Process

Data were extracted from each article based on a pre-established summary table. The table included the following information: author, year of publication, sample characteristics (sample number, sex, and age), dose and duration of treatment with FA supplementation, serum concentrations of folate, folate status, and FA food fortification policies. Two reviewers (Zhang. L. and Chen. X.) independently extracted all the data, and any discrepancies were resolved by consensus. 

### 2.7. Study Risk of Bias Assessment

Two reviewers (Zhang. L and Chen. X) independently assessed the risk of bias in individual studies. In case of any discrepancies, resolved them by consensus. If required, they consulted another reviewer (Li. W). Each RCT used Review Manager 5.4 to evaluate the study quality according to the Cochrane tool for assessing risk of bias [11]. Funnel plots and Egger’s test were developed using Stata MP 17 to assess the probability of publication bias. 

### 2.8. Statistical Analysis

The meta-analysis utilized Cochrane Review Manager 5.4 and Stata MP 17 to integrate data with the same outcome but measured with different scales. 

Due to the variation in participants, intervention doses, the duration of interventions, and baseline folate levels, there was a high degree of heterogeneity in the data. As a result, the standardized mean difference (SMD) and its 95% confidence interval (CI) of the global cognitive function change scores in the intervention and control groups in the pre- and post-intervention periods are presented. When the standard deviation (SD) of mean change (SD_change_) is not presented, we have calculated the SD change with the following formula: SD_change_ = $\sqrt{\frac{(n_{1}-1)\;{S_{1}}^{2}+(n_{2}-1)\;{S_{2}}^{2}}{n_{1}+n_{2}-2}}$. *S*_1_ and *S*_2_ represent the SD before and after FA was used for intervention.

We performed a subgroup analysis of included data based on the characteristics of mandatory food FA fortification at the beginning of the study. The aim was to evaluate the effectiveness of the intervention in areas where different FA fortification policies were implemented. We used https://fortificationdata.org/map-number-of-food-vehicles/ (accessed on 21 June 2023) to determine whether each country implemented mandatory food FA fortification and to identify the types of fortified foods and the date of implementation. We compared the timing of the trial intervention with the timing of the mandatory food FA fortification and categorized the records into two categories: fortified and unfortified. Considering that cognitive change is a slow progress, we have stratified the analysis based on the intervention duration to ensure the stability of the analysis results.

## 3. Results

### 3.1. Study Selection

The process of selecting the studies is presented in Figure 1, which shows the flowchart for identifying the studies. Initially, 817 records were identified by searching various databases and by manual reverse search. After removing the duplicate records, 314 records were left. Two hundred sixty-two records were excluded as they needed to meet the inclusion criteria after reading the title and abstract. Later, ten additional manuscripts were retrieved by manually searching other published reviews and articles. After carefully reading the 62 retained articles, 11 articles were removed, including 4 studies from the same team [12,13,14,15], 1 article did not perform a cognitive measure [16], three articles reanalyzed other articles [17,18,19], and 3 research studies used neuroimaging techniques to measure variation in outcomes [20,21,22]. Finally, the analysis included the final 51 articles.

### 3.2. Characteristics of Individual Trials 

We have thoroughly examined the 51 records that meet the inclusion criteria and have recorded detailed characteristics of the studies in Table 2. The 51 RCTs included a total of 42,768 participants, aged between 60 and 80 years. The duration of treatment varied from 1 month to 8.5 years across the different trials. In all the trials, the treatment with FA was compared to the control treatment, or the treatment with FA, vitamin B_12,_ or vitamin B_6_ (or both) was compared to the control treatment.

### 3.3. Effect of Folate-Based B Vitamin Supplementation on Cognitive Performance of Older Adults Based on Mandatory FA Fortification

The selected trials were categorized into two groups based on mandatory FA fortification: fortified and no-fortified. The fortified group consisted of twelve articles, of which three reported an effective intervention (25.00%), three reported a reduction in homocysteine levels, and six reported a non-significant effect of the folate-based B vitamin intervention [23,24,25,26,27,28,29,30,31,32,33,34]. The no-fortified group comprised 39 articles, mostly from studies in China and the United Kingdom, with 16 articles (48.72%) reporting that the intervention was effective [35,36,37,38,39,40,41,42,43,44,45,46,47,48,49,50,51,52,53,54,55,56,57,58,59,60,61,62,63,64,65,66,67,68,69,70,71,72,73]. 

Providing folate-based B vitamin to people with cognitive impairment (MCI/AD) has shown mostly effective results. Three articles studied the MCI/AD population in the fortified group, and one had a partial valid result (33.33%). Eighteen articles studied the MCI/AD population in the no-fortified group, with fifteen being valid for the outcome (including lowering homocysteine levels), with an efficiency rate of 83.33%. 

### 3.4. Meta-Analysis of Folate-Based B Vitamin on Cognitive Performance of Older Adults 

The different trials’ focus of interest was the alteration in cognitive ability in different cognitive domains, which were evaluated using several cognitive scales from the beginning of study until the final follow-up. This meta-analysis focused on the changes in cognitive ability, so studies that did not report the changes in cognitive ability could not be combined in the meta-analysis. Therefore, only 23 articles were considered eligible for the meta-analysis. 

Figure 2 showed the results of the meta-analysis of the effects of folate-based B vitamin supplementation on changes in cognitive function scores in older adults across 23 trials. Data extracted from 23 RCTs and random-effects model results showed a significant effect of the intervention group versus the control group on global cognitive function (SMD: 0.18; 95% CI: 0.09–0.28, *p* = 0.0001). Subgroup analysis revealed that the FA intervention’s effect was insignificant in the fortified area (SMD: 0.03; 95% CI: −0.06 to 0.13, *p* = 0.45). However, there were significant differences between the intervention and control groups in the unfortified area (SMD: 0.27; 95% CI: 0.13 to 0.40, *p* = 0.0002). There was no overlap in the 95% CI for SMD between the fortified and no-fortified groups, and the difference between subgroups was statistically significant (*p* = 0.007), indicating a significant interaction between grouping and combined effect size.

As cognitive change is a gradual process, we divided the data based on the length of the intervention. After excluding studies with less than one year of folate-based B vitamin intervention, the analyses we conducted showed a significant difference between the intervention group and the control group in terms of global cognitive function (SMD: 0.13; 95% CI: 0.03 to 0.23, *p* = 0.01). Further analysis revealed that the effect of folate-based B vitamin intervention was insignificant in the fortified area (SMD: 0.03; 95% CI: −0.06 to 0.13, *p* = 0.45). However, there were significant differences between the intervention and control groups in the unfortified area (SMD: 0.22; 95% CI: 0.04 to 0.40, *p* = 0.01) (Figure 3A). When we reanalyzed studies that excluded folate-based B vitamin interventions for less than two years, we found similar results to those excluding less than 12 months (SMD: 0.13; 95% CI: 0.02 to 0.24, *p* = 0.02). The same results were found in subgroup analyses, in the fortified area (SMD: 0.03; 95% CI: −0.08 to 0.13, *p* = 0.60), and the unfortified area (SMD: 0.22; 95% CI: 0.04 to 0.40, *p* = 0.01) (Figure 3B). 

In areas without mandatory FA food fortification, folate-based B vitamin interventions were associated with a trend of subsequent cognitive improvement, regardless of whether studies with less than one or two years of intervention were excluded. To sum up, it is evident that the duration of the intervention has no impact on the effectiveness of the folate-based B vitamin intervention.

### 3.5. Risk of Bias within Studies

All 51 studies were assessed individually using the Cochrane Risk of Bias tool. The summary plots of risk of bias show that most studies had a low risk of reporting and other biases (Appendix A). Appendix A literates the risk of bias for the literature included in the meta-analysis. Two studies each were given a high-risk rating for selection bias, performance bias, and detection bias. In Appendix A, the funnel plot showed that most studies are within the triangle, with a few individual studies outside the triangle. Also, the results of Egger’s regression test showed that *t* = 2.82, *p* < 0.05, all indicating some publication bias.

## 4. Discussion

This study analyzed fifty-one published studies on the effect of folate-based B vitamin supplementation and cognitive function in older adults. The intervention efficiency of FA fortification was evaluated in two groups: areas with or without mandatory policy. The intervention efficiency of the fortified group was 25.00%, while the no-fortified group had an efficiency of 48.72%, which is higher. A meta-analysis was then conducted, which revealed that folate-based B vitamin supplementation improved cognitive function in older adults in countries without mandatory food FA fortification. However, there was no significant improvement in cognitive function in older adults in areas with mandatory fortification. The stratified analysis of intervention times yielded consistent results. This conclusion aligns with previous results on the effects of folate-based B vitamin supplementation on cardiovascular disease, which demonstrated no benefit in stroke prevention in areas where FA fortification policies are established [74]. Therefore, food fortification with FA plays a crucial role in determining the effectiveness of folate-based B vitamin interventions. 

The fortification of wheat or corn flour with FA, either alone or in combination with other micronutrients, has increased red blood cell and serum/plasma folate concentrations [75]. As a result, there has been a decline in the rate of folate deficiency in the United States [76]. High serum folate has also been found to prevent cognitive impairment, according to a cross-sectional analysis in a National Health and Nutrition Examination Survey study in the United States [77]. 

However, it is essential to note that folate is a water-soluble vitamin, and studies have shown that supplementation with FA in individuals who already have adequate folate levels only results in saturation. Any extra FA in such individuals is excreted in the urine [78], and as the intake of folate increases, so does the amount excreted in the urine [79]. While dietary folate and some supplement FA are transformed into 5-methyltetrahydrofolate, which is the active form of folate in the body, the remaining supplement FA cannot be absorbed and becomes unmetabolized folic acid (UMFA), which cannot function [80]. 

Data from a cross-sectional nutrition survey in the United States has shown that chronic exposure to FA in physiologic doses (as would be the case with mandatory fortification) might induce saturation, leading to more UMFAs [81]. Therefore, consuming more FA than the body needs plays a minor role.

These findings suggest that the higher intake of FA through fortified foods has reduced the potential therapeutic effect of the active treatment group. Therefore, excessive FA supplementation does not provide any additional health benefits. However, countries without mandatory food FA fortification in their food supply have lower levels of FA and high folate deficiency rates, indicating a significant impact of FA replenishment [57].

The connection between B vitamins and cognitive function has been a subject of interest for a long time. Various systematic reviews and meta-analyses have utilized different methods to answer this question. In a previous study, we found that the folate level at baseline could impact the effect of FA supplementation on cognitive function in older adults and that those deficient in folate would benefit more from supplementation [82]. In this study, we further investigated the FA levels of older adults with and without fortification, focusing on differences in the effects of interventions on population living in areas with varying FA levels. In addition, the studies included in this review were numerous and were published over an extended period. Two other articles used a similar approach to our analysis of results for global cognitive function, and both concluded that there was no significant cognitive benefit from vitamin B supplementation [1,83]. However, there were differences in our calculations and the number of articles included. A meta-analysis by Zhang et al. examined the relationship between vitamin B_12_, B_6_, and FA intake levels and cognitive function in an elderly population and found no significant benefit on cognitive function. The most apparent difference between this review and ours is that it is based on cohort study, whereas our systematic evaluation was based on an RCT study [84].

After implementing FA fortification policy, the study analyzed FA intervention trials. The folate-fortified group of this study included the United States and Canada (since 1998) and Australia (since 2009). In our analysis, only countries that queried explicit mandatory food fortification policies were categorized into fortification groups. Studies with an unclear implementation of food fortification policies were included in the non-mandatory food fortification group. These countries include France, the Netherlands, Japan, Sweden, Singapore, and Germany.

There are a few special articles that need to be explained here. Two of these articles, by Harris and Macpherson, used multivitamins, minerals, and herbs to intervene with older adults, and both had positive results [27,30]. Another article examined FA’s potential to prevent stroke recurrence by lowering homocysteine levels in stroke patients. Although cognition was not a primary outcome, the study measured it with MMSE before and after the intervention [33]. Lastly, the study excluded an RCT by Shah et al. that examined the effectiveness of medical foods containing FA on cognitive performance in patients with Alzheimer’s because the specific dose of FA used was not available [85].

## 5. Limitations

As with most systematic reviews and meta-analyses, our article has several limitations. First, the interpretation of our findings needs to be completed because we only considered RCTs and not cohort and case–control studies. To minimize bias, we included more RCTs and did not limit the literature’s publication time. Second, the age range of the study population was broad, so we limited the age of the study population in the literature to 60 years and older to reduce the heterogeneity. Third, we only analyzed the full range of cognitive functions, which resulted a small number of trials being included in the meta-analysis. Fourth, all articles of the fortified area included in the meta-analysis used a mixed B vitamin intervention with folic acid as the primary ingredient, so this study was unable to investigate the effectiveness of the folic acid alone as an intervention. Fifthly, studies from different countries and regions have varied in intervention duration, reinforcement dose, baseline folate status, and adjustment for confounders, potentially leading to heterogeneity. Lastly, there is a possibility that publication bias can affect the overall findings of the meta-analysis. Therefore, it is essential to interpret the results of this review with caution. In future studies, using consistent and generalized cognitive functioning would be beneficial, ensuring the better integrated analysis of the results from different studies.

## 6. Conclusions

Based on the data analyzed in this meta-analysis, it was found that folate-based B vitamin supplementation can significantly improve cognitive function in older adults, especially in countries where mandatory food FA fortification is not practiced. However, no significant impact of folate-based B vitamin supplementation was observed in areas where FA fortification in foods was mandatory. This suggests that mandatory food fortification may mask the effects of folate-based B vitamin supplementation on cognitive function. In other words, additional nutritional supplements may provide less benefit than the mandatory fortification of foods. 

It is important to note that older adults in areas without mandatory FA fortification may experience severe folate deficiency. Thus, the advantages of FA supplementation can be substantial and have considerable public health benefits in such areas. 

## Figures and Tables

**Figure 1 nutrients-16-02199-f001:**
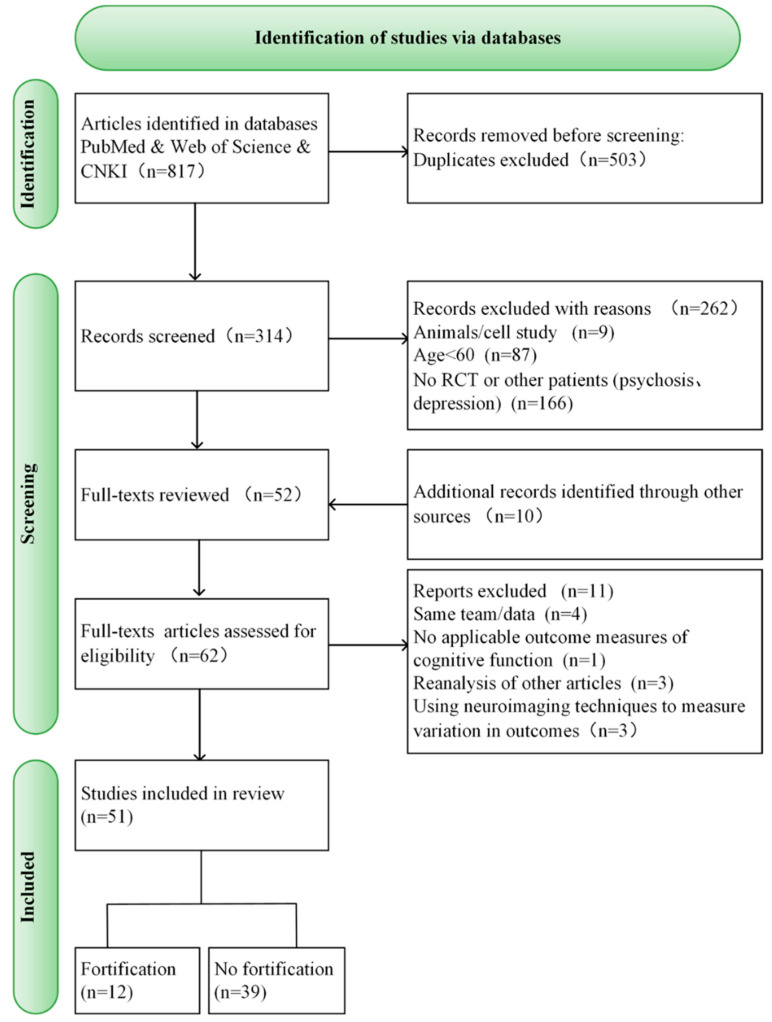
Flowchart summary of search process.

**Figure 2 nutrients-16-02199-f002:**
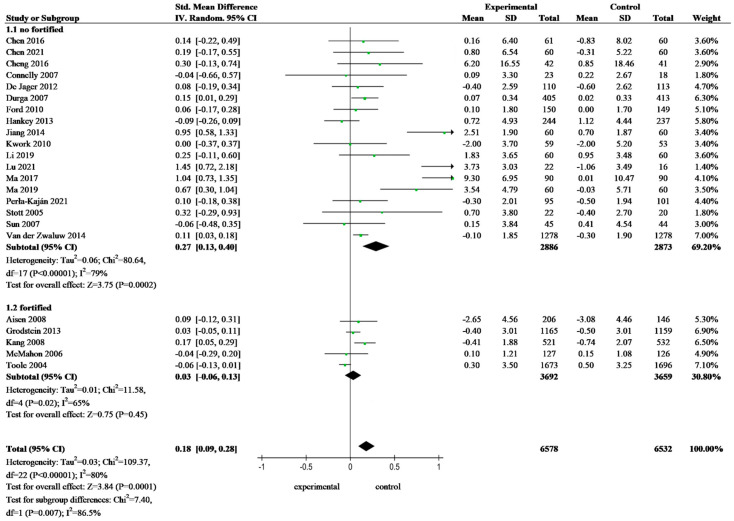
Forest plots showing standardized mean difference on the global cognition function of older adults—subgroup analysis SMD, standardized mean difference; CI, confidence interval, [23,26,28,29,33,39,40,41,42,43,44,47,50,51,52,55,56,59,60,62,65,66,71].

**Figure 3 nutrients-16-02199-f003:**
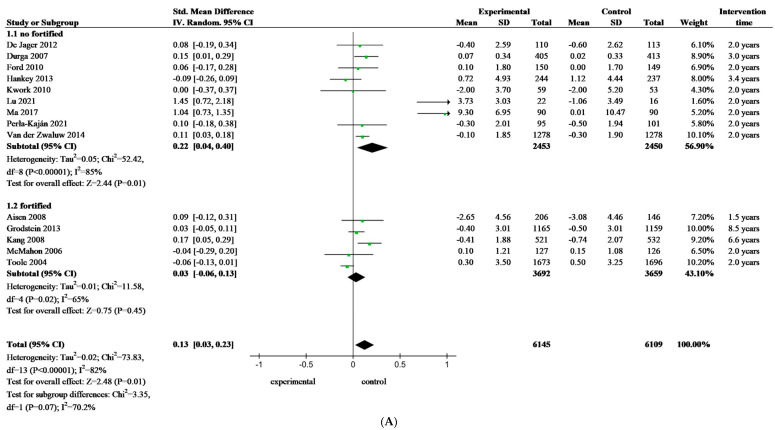
Forest plots of standardized mean difference on the global cognition function of older adults—subgroup analysis. (**A**): excluding studies with less than 12 months, [23,26,28,29,33,43,44,47,50,52,56,59,62,71]. (**B**): excluding studies with less than 24 months. SMD, standardized mean difference; CI, confidence interval, [26,28,29,33,43,44,47,50,52,56,59,62,71].

**Table 1 nutrients-16-02199-t001:** PICOS eligibility criteria for inclusion.

	Included in Review	Excluded from Review
Patient population	Senior citizens with an average or median age of over 60 years	Pre-existing mental disorders Children and adolescents Pregnant women
Intervention	FA supplementation either alone or in conjunction with B_6_/B_12_ or other micro-nutrients	Additional interventions other than stated on the left The use of vitamin B was not specified
Comparison	Control group (placebo)	Any other
Outcomes	cognitive outcome measures	Non-cognitive outcome measures Studies lacking data on change of cognitive measures
Study design	RCTs (randomized controlled trials)	Not RCT

**Table 2 nutrients-16-02199-t002:** Study details and sample demographics.

Fortification	Study and Year	Country	Participants	Age (y) ^e^	Female (%)	Folate Concentration at Baseline	Folate Status	Supplement			Duration of Treatment	Efficiency	Fortification Food
								Folic Acid	Vitamin B12	Vitamin B6			
Yes	Aisen et al., 2008 [23]	USA	409 individuals with mild to moderate AD	I: 75.70 ± 8.00 C: 77.30 ± 7.90	55.99	I: 25.74 ± 17.65 nmol/L C: 23.59 ± 14.35 nmol/L	Fully ^a^	Yes	Yes	Yes	1.5 years	Partly effective ^d^	Maize flour; rice; wheat flour
	Brady et al., 2009 [24]	USA	659 elders	I: 63.20 ± 12.20 C: 64.20 ± 11.20	1.67	I: 15.9 ng/mL C:14.8 ng/mL	Fully ^a^	Yes	Yes	Yes	5 years	Ineffective	
	Chan et al., 2010 [25]	USA	584 elders	No details	No Data	No details	No details	Yes	Yes	No	either 2 weeks or 3 months	Effective	
	Grodstein et al., 2013 [26]	USA	5947 elders	I: 71.60 ± 6.00 C: 71.60 ± 5.90	0	No details	No details	Yes	Yes	Yes	8.5 years	Ineffective	
	Harris et al., 2012 [27]	Australia	51 elders	I: 62.00 ± 3.90 C: 62.40 ± 6.60	0	I: 575.0 ± 228.1 ng/mL C: 533.0± 209.8 ng/mL	Fully ^a^	Yes	Yes	Yes	2 months	Effective	Wheat flour
	Kang et al., 2008 [28]	USA	5442 elders	I: 71.30 ± 4.20 C: 71.30 ± 4.20	100	No details	No details	Yes	Yes	Yes	6.6 years	Ineffective	
	McMahon et al., 2006 [29]	New Zealand	276 elders	I: 73.60 ± 5.80 C: 73.40 ± 5.70	40.58	I: 10.0 ± 5.0 ng/mL C: 10.0 ± 5.0 ng/mL	Fully ^a^	Yes	Yes	Yes	2 years	Partly effective ^d^	Wheat flour
	Macpherson et al., 2012 [30]	Australia	56 elders	I: 71.90 ± 4.80 C: 70.30 ± 4.30	100	No details	No details	Yes	Yes	Yes	4 months	Partly effective ^d^	
	Rommer et al., 2015 [31]	Australia	48 patients with MCI/AD	I: 76.40 ± 6.70 C: 63.30 ± 13.70	39.58	No details	No details	Yes	Yes	Yes	3 months	Ineffective	
	Sommer et al., 2003 [32]	USA	11 patients with AD	T: 76.70 ± 4.10	42.86	No details	No details	Yes	No	No	10 weeks	Ineffective	
	Toole et al., 2004 [33]	USA, Canada and Scotland	3680 elders	I: 66.40 ± 10.80 C: 66.20 ± 10.80	37.40	No details	No details	Yes	Yes	Yes	2 years	Ineffective	
	Walker et al., 2012 [34]	Australian	900 elders	I: 65.92 ± 4.30 C: 65.97 ± 4.18	60.22	I: 572.54 ± 266.32 nmol/L C: 557.09 ± 277.50 nmol/L	Fully ^a^	Yes	Yes	No	2 years	Effective	
No	Andreeva et al., 2011 [35]	France	871 elders	I: 61.40 ± 8.70 C: 60.90 ± 8.90	21.70	I: 6.90 ± 3.50 ng/mL C: 7.00± 3.70 ng/mL	Insufficiency ^a^	Yes	Yes	Yes	4 years	Ineffective	
	Bryan et al., 2002 [36]	Australia	75 elders	T:74.08 ± 5.75	100	No details	No details	Yes	Yes	Yes	5 weeks	Effective	
	Cockle et al., 2000 [37]	UK	139 healthy elders	I: 70.70 ± 5.60 C: 70.20 ± 5.40	63.31	No details	No details	Yes	Yes	Yes	6 months	Ineffective	
	Clarke et al., 2003 [38]	UK	149 patients with MCI/AD	T: 75.00	No Data	T: 7.1 nmol/L	Insufficiency ^a^	Yes	Yes	No	3 months	Partly effective ^d^	
	Connelly et al., 2007 [39]	UK	57 patients with AD	I: 75.65 ± 5.94 C: 77.60 ± 6.89	50.88	I: 9.77 ± 5.66 mg/L C: 8.71 ± 4.54 mg/L	Fully ^a^	Yes	No	No	6 months	Ineffective	
	Chen et al., 2016 [40]	China	121 patients with AD	I: 68.10 ± 8.50 C: 67.63 ± 7.92	50.41	I: 11.98 (8.04–17.68) nmol/L C: 10.76(7.53–16.37) nmol/L	Insufficiency ^b^	Yes	No	No	6 months	Effective	
	Cheng et al., 2016 [41]	China	104 healthy elders	I: 74.30 ± 9.60 C: 72.50 ± 7.00	51.92	I: 9.48 ± 5.2 ng/mL C: 9.90 ± 4.0 ng/mL	Fully ^a^	Yes	Yes	Yes	14 weeks	Effective	
	Chen et al., 2021 [42]	China	120 patients with AD	I: 68.58 ± 7.29 C: 68.02 ± 8.34	46.67	I: 14.37 (9.52–19.32) nmol/L C: 16.08(11.35–24.58) nmol/L	Insufficiency ^a^	Yes	Yes	No	6 months	Effective	
	De Jager et al., 2012 [43]	UK	266 patients with MCI	I: 76.80 ± 5.10 C: 76.70 ± 4.80	64.13	I: 22.6 (20.0–25.5) nmol/L C: 23.0 (20.4–26.0) nmol/L	Fully ^c^	Yes	Yes	Yes	2 years	Effective	
	Durga et al., 2007 [44]	The Netherlands	818 elders	I: 60.0 ± 5.0 C: 60.0 ± 6.0	28.36	I: 12 (9–15) nmol/L C: 12 (10–15) nmol/L	Insufficiency ^a^	Yes	No	No	3 years	Effective	
	Eussen et al., 2006 [45]	The Netherlands	195 elders	I: 83 ± 6 C: 82 ± 5	76.41	I: 591 ± 203 nmol/L C: 680 ± 280 nmol/L	Fully ^a^	Yes	Yes	No	6 months	Ineffective	
	Fioravanti et al., 1997 [46]	Italy	30 healthy elders	I: 80.25 ± 5.78 C: 80.21 ± 5.45	83.33	I: 2.34 ± 0.51 ng/mL C: 2.21 ± 0.68 ng/mL	Insufficiency ^a^	Yes	No	No	2 months	Effective	
	Ford et al., 2010 [47]	Australia	299 healthy elders	I: 79.30 ± 2.80 C: 78.70 ± 2.70	0	I: 24.00 ± 0.61 nmol/L C: 24.40 ± 0.61 nmol/L	Fully ^c^	Yes	Yes	Yes	2 years	Ineffective	
	Gong et al., 2022 [48]	Hongkong	279 patients with MCI	I: 76.90 ± 5.40 C: 76.72 ± 5.22	41.86	I: 28.26 ± 8.06 nmol/L C: 28.82 ± 8.65 nmol/L	Fully ^a^	Yes	Yes	No	2 years	Effective	
	Hama et al., 2020 [49]	Japan	45 elders	T: 79.7 ± 7.9	62.2	No details	Insufficiency ^b^	Yes	No	No	28–63 days	Effective	
	Hankey et al., 2013 [50]	Australia	2214 elders	T: 63.60 ± 11.80	32.7	No details	No details	Yes	Yes	Yes	3.4 years	Partly effective ^d^	
	Jiang et al., 2014 [51]	China	120 healthy elders	T: 63.00 ± 1.90	35	I: 2.74 ± 0.65 ng/mL C: 2.83 ± 0.80 ng/mL	Insufficiency ^a^	Yes	Yes	No	6 months	Effective	
	Kwok et al., 2010 [52]	Hongkong	140 patients with AD	I: 79.10 ± 6.70 C: 77.20 ± 7.90	63.57	I: 21.70 ± 9.10 nmol/L C: 20.0 ± 7.0 nmol/L	Fully ^a^	Yes	Yes	No	2 years	Ineffective	
	Kwok et al., 2019 [53]	Hongkong	279 patients with MCI	I: 77.80 ± 5.55 C: 78.00 ± 5.30	40.5	I: 27.80 ± 8.00 nmol/L C: 29.40 ± 8.60 nmol/L	Fully ^a^	Yes	Yes	No	2 years	Partly effective ^d^	
	Lewerin et al., 2005 [54]	Sweden	209 healthy elders	I: 75.70 ± 4.70 C: 75.60 ± 4.00	55.90	I: 15.7 ± 6.1 nmol/L C: 16.4 ± 5.1 nmol/L	Insufficiency ^a^	Yes	Yes	Yes	4 months	Ineffective	
	Li et al., 2019 [55]	China	240 patients with MCI	I: 70.20 ± 6.13 C: 70.38 ± 6.73	58.75	I: 8.41 ± 4.15 ng/mL C: 7.99 ± 4.84 ng/mL	Insufficiency ^a^	Yes	No	No	6 months	Effective	
	Lu et al., 2021 [56]	China	50 healthy elders	I: 66.16 ± 7.61 C: 69.00 ± 10.80	74	I: 7.96 ± 2.10 ng/mL C: 8.77 ± 4.05 ng/mL	Fully ^a^	Yes	Yes	No	2 years	Effective	
	Ma et al., 2015 [57]	China	180 patients with MCI	I: 74.82 ± 2.75 C: 74.63 ± 3.21	57.22	I: 7.01 ± 3.64 ng/mL C: 5.79 ± 2.67 ng/mL	Insufficiency ^b^	Yes	No	No	6 months	Effective	
	Ma et al., 2016 [58]	China	168 patients with MCI	I: 73.71 ± 2.57 C: 73.52 ± 3.03	68.45	I: 7.01 ± 1.01 ng/mL C: 6.33 ± 0.97 ng/mL	Fully ^a^	Yes	No	No	1 years	Effective	
	Ma et al., 2017 [59]	China	180 patients with MCI	I: 74.82 ± 2.75 C: 74.63 ± 3.21	57.22	I: 7.01 ± 0.64 ng/mL C: 5.79 ± 0.67 ng/mL	Insufficiency ^b^	Yes	No	No	2 years	Effective	
	Ma et al., 2019 [60]	China	240 patients with MCI	I: 68.42 ± 3.62 C: 68.54 ± 3.90	64.17	I: 7.71 ± 1.17 ng/mL C: 7.61 ± 0.60 ng/mL	Insufficiency ^b^	Yes	Yes	No	6 months	Effective	
	Pathansali et al., 2005 [61]	UK	24 healthy elders	I: 72.30 ± 6.00 C: 73.80 ± 5.30	87.50	I: 6.0 ± 2.3 μg/L C: 6.6 ± 2.8 μg/L	Insufficiency ^a^	Yes	No	No	1 months	Ineffective	
	Perła-Kaján et al., 2021 [62]	UK	196 patients with MCI	T: 77.60 ± 4.80	60	No details	No details	Yes	Yes	Yes	2 years	Effective	
	Remington et al., 2014 [63]	New England	106 patients with AD	T: 77.80± 9.30	No Data	No details	No details	Yes	Yes	No	9 months	Effective	
	Remington et al., 2015 [64]	New England	34 patients with MCI	T: 65.90 ± 11.30	No Data	No details	No details	Yes	Yes	No	1 years	Effective	
	Stott, 2005 [65]	UK	185 elders	I: 72.60 ± 6.40 C: 72.80 ± 5.40	44.68	I: 320 ± 122 ng/mL C: 269 ± 87 ng/mL	Fully ^a^	Yes	Yes	Yes	3 months	Partly effective ^d^	
	Sun et al., 2007 [66]	Taiwan	89 patients with AD	I: 74.90 ± 7.10 C: 74.60 ± 7.50	49.44	I: 9.0 ± 4.50 ng/mL C: 8.40 ± 6.60 ng/mL	Insufficiency ^a^	Yes	Yes	Yes	26 week	Partly effective ^d^	
	Search et al., 2010 [67]	UK	12064 elders	T: 64.20± 8.90	17.01	No details	No details	Yes	Yes	No	6.7 years	Partly effective ^d^	
	Tan et al., 2022 [68]	Singapore	707 elders	I: 61.51 ± 11.28 C: 60.22 ± 11.47	31.82	I: 17.10 ± 8.32 μmol/L C: 16.62 ± 7.77 μmol/L	Fully ^a^	Yes	Yes	Yes	5 years	Ineffective	
	Ting et al., 2017 [69]	Singapore	230 elders	I: 68.00 C: 66.00	39.57	No details	No details	Yes	Yes	Yes	5 years	Partly effective ^d^	
	van Uffelen et al., 2008 [70]	The Netherlands	152 patients with MCI	I: 75.40 ± 2.80 C: 74.90 ± 3.00	44.08	No details	No details	Yes	Yes	Yes	1 years	Ineffective	
	van der Zwaluw et al., 2014 [71]	The Netherlands	2919 elders	T: 74.1 ± 6.5	50.00	I: 19.2 (14.0–22.6) nmol/L C: 18.9 (14.2–22.7) nmol/L	Fully ^a^	Yes	Yes	No	2 years	Partly effective ^d^	
	van Soest et al., 2022 [72]	The Netherlands	191 elders	I: 70.30 ± 5.10 C: 72.70 ± 6.30	44	I: 16.9 (13.9–22.4) nmol/L C: 17.7 (14.3–24.7) nmol/L	Fully ^a^	Yes	Yes	No	2 years	Effective	
	Wolters et al., 2005 [73]	Germany	220 elders	I: 63.00 C: 64.00	100	No details	No details	Yes	Yes	Yes	6 months		

Unit conversion formula for serum folate: 1 ng/mL ≈ 2.265 nmol/L. ^a^ Calculation based on WHO definitions of folate sufficiency [serum (plasm) folate below 10 nmol/L (4 ng/mL); red blood cell folate below 151 ng/mL (340 nmol/L)]; ^b^ Results came from original studies; ^c^ Be consistent with the primary study; ^d^ Partly effective (reducing homocysteine); ^e^ Data are shown as mean ± SD. Abbreviation: I: intervention group; C: control group; T: total; AD: Alzheimer’s disease; MCI: mild cognitive impairment.

## Data Availability

All of the data generated or analyzed during this review are included in this published article.

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
