# Peer review of "A Comparative Study Evaluating the Effectiveness of Folate-Based B Vitamin Intervention on Cognitive Function of Older Adults under Mandatory Folic Acid Fortification Policy: A Systematic Review and Meta-Analysis of Randomized Controlled Trials"

_nutrients, 2024, doi:10.3390/nu16142199_

Round 1

Reviewer 1 Report

Comments and Suggestions for Authors

Introduction should be improved. The reader should be provided with background information, and the relevance of B vitamin supplementation, especially folic acid 45 (FA), to prevent cognitive impairment in adults should be indicated and justified. As well as, what the current scientific literature says about the state of the population in this respect etc. In other words, they should put the reader in the background.

About methods, It is not clear how publication bias was analysed.

Comments on the Quality of English Language

Minor editing of English language required.

Reviewer 2 Report

Comments and Suggestions for Authors

The study effectively outlines the study's purpose, methodology, and findings. It provides a clear overview of the research process, including the databases searched, inclusion criteria, and analysis methods. 

Please complete the following comments

1. The inclusion of studies from various regions and populations may introduce heterogeneity, which can affect the generalizability of the findings

2. The review assumes a binary classification of countries based on FA fortification policies (mandatory vs. non-mandatory). However, the implementation and compliance levels of these policies may vary, potentially influencing the outcomes.

3. There is a possibility of publication bias, where studies with positive results are more likely to be published. This can skew the overall findings of the meta-analysis.

4. Cognitive function changes over time, and the included studies may have varied follow-up durations. Long-term effects of supplementation might not be adequately captured in studies with shorter follow-up periods.

5. The baseline nutritional status of participants was not consistently accounted for across studies. Pre-existing levels of folate and other B vitamins could influence the efficacy of supplementation.

6. Cognitive function in older adults can be influenced by numerous factors, including lifestyle, comorbidities, and other medications. The included studies may not have fully accounted for these potential confounders.

By addressing these limitations this review could be significant conclusion
